# OpenReview forum: "Learning Representations of Instruments for Partial Identification of Treatment Effects"
_ICML.cc/2025/Conference — ICML 2025 poster_

### Official Review · Reviewer_xPaP · 2025-02-24

**Overall Recommendation:** 3

**Summary:**

The paper presents a method for learning bounds on CATE (conditional average treatment effect) in the event of observed covariates X, unobserved confounding U, binary treatment A, scalar outcome Y,  and an instrument Z which may be high dimnensional.

The proposed method extends:

Schweisthal, J., Frauen, D., van der Schaar, M., and Feuer- riegel, S. Meta-learners for partially-identified treatment effects across multiple environments. In ICML, 2024.

which deals with the case of discrete instruments, and provides the core result of Lemma 2. The contribution of the paper is a neural net approach to mapping the high dimensional continuous variable Z to a set of k partitions. Consequent bounds on the CATE are then found straightforwardly (Theorem 1).

**Claims And Evidence:**

This paper presents a reasonable approach to a setting with some real-world application, and has some methodological improvements over the prior work by Schweisthal et al. that it extends. However, the contribution might be considered incremental.

**Essential References Not Discussed:**

n/a

**Experimental Designs Or Analyses:**

see above

**Methods And Evaluation Criteria:**

A number of strategies are employed to train the instrument discretization network. These combine: 1) minimizing the bound width, 2) ensuring no bin has too small a mass, 3) enforcing that the distribution of Z given its bin is heterogeneous across bins.  These approaches are all reasonable so I have no questions or objections.

The evaluation is conducted on three synthetic benchmarks, to ensure a known ground truth. Since no method yet addresses the setting, the main point of comparis on is the naive k-means discretization on the one hand, as well as the known ground-truth CATE.  Again, this choice is reasonable, and I have no questions on it.

**Other Comments Or Suggestions:**

n/a

**Other Strengths And Weaknesses:**

n/a

**Questions For Authors:**

n/a

**Relation To Broader Scientific Literature:**

see above

**Theoretical Claims:**

Asumptotic normality of the conditional mean and propensity score estimates is shown by combining CLT and delta method (Theorem 2).

I did not read the proofs in detail but standard techniques are used.

---

> ### Author Rebuttal · Authors · 2025-04-01
>
> Thank you a lot for your positive feedback and your comments! We are happy to see that our claims, methods, theory, and experiments are well received and do not raise any additional concerns.
>
> Here, we would kindly like to elaborate on why our paper is **not** an incremental contribution of the work of Schweisthal et al. but instead a standalone work with multiple novelties in a different setting.
>
> (i) **Applicability to arbitrary IVs and target bound width minimization**: We show that the existing bounds for discrete instruments from Lemma 1 (which Schweisthal et al. used and which are also just an adaptation of the Manski bounds) can be applied to other instrument types (continuous, high-dimensional) by using _arbitrary partitioning functions_, enabling to transfer and generalize the bounds to new unconsidered settings such as Mendelian randomization (MR) or indirect experiments with complex nudges. **This is a novel theoretical finding**. While this may seem straightforward at first sight, to the best of our knowledge, we are not aware of any prior work considering that connection, i.e., _even our naive baseline leveraging $k$-means clustering has not been considered before_.
>
> This is orthogonal to the work of Schweisthal et al. who derived model-agnostic learners in the **discrete** IV setting, while we focus on **complex and continuous** IVs. Further, this finding allows us to develop the **new objective of directly targeting bound width minimization** during representation learning to learn optimal partitions (Eq. (8)). Based on this, **we  make two major theoretical contributions regarding optimized training**:
>
> (ii) **Stability by avoiding alternating learning**: A straightforward implementation minimizing the bounds following Eq. (8) would require alternating learning. The reason is that, after every update step of $\phi(z)$, the quantities $\mu_\phi^a(x, \ell)$ and $\pi_\phi(x, \ell)$ are not valid for the updated $\phi$ anymore and would need to be retrained to ensure valid bounds. This is computationally highly expensive and results in unstable training and convergence problems. However, our method circumvents these issues: by using our novel Theorem 1, we show that, while training $\phi(z)$, the quantities $\mu_\phi^a(x, \ell)$ and $\pi_\phi(x, \ell)$ can be directly calculated. (see also our subsection “Implications of Theorem 1” on page 4).
>
> For that, we can simply evaluate the nuisance functions, which only need to be trained once in the first stage. Therefore, we avoid any need for alternating learning, resulting in more efficient and stable training. Here, also note that Theorem 1 and its proof in Appendix A1 target effective _estimation_ of our target quantities, and **thus is orthogonal to the works about discrete instruments** which aim for the _derivation_ of bounds.
>
> (iii) **Improved finite sample robustness**: Even using our stable training procedure from above, optimizing for Eq. (4) only yields valid bounds _in expectation on the population level_. However, if the discrete representation learning results in highly imbalanced marginal probabilities during training (i.e., $\mathbb{P}(\phi(Z)=\ell)$ is small for some $\ell$), this can result in high estimation variance of the nuisance estimates and thus unreliable bound estimates. **We show this more formally in our Theorem 2 where we provide theoretical guarantees for the asymptotic behavior**.
>
>  In contrast, we avoid these problems: by using our theoretically motivated custom loss from Eq. 19 with the respective regularization from Eq. 17, _we enforce lower estimation variance during training and thus more reliable bound estimates_.
>
> In sum, we only leverage the formulation of the closed-form bounds of Schweisthal et al. from the discrete IV setting ( - which is also **not** their main contribution but only extending Manski bounds to the CATE - ) as a simple starting point for our method. Thus _our major contributions are independent of the contributions of Schweisthal et al_., which are model-agnostic meta-learners in a different setting (discrete vs. complex continuous IVs).
>
> **Action**: We further improved the comparison to previous work in our paper to clarify the novelty of our method.

---

> > ### Comment · Reviewer_xPaP · 2025-04-07
> >
> > Thank you for the rebuttal. I think the explanations above add clarity to the paper, and would be helpful to incorporate in a final version. I will maintain the current score.

---

> > > ### Author Response · Authors · 2025-04-08
> > >
> > > Dear Reviewer xPaP,
> > >
> > >
> > > Thank you for your response and your positive evaluation of our work! As promised, we will incorporate our explanations and the more direct comparison with prior work from above into our paper to improve the clarity, in particular with regard to the novelty of our method.
> > >
> > > Best regards,
> > >
> > > The Authors

---

### Official Review · Reviewer_m2Ds · 2025-03-14

**Overall Recommendation:** 3

**Summary:**

This paper proposes a new method for partial identification of the conditional treatment effect (CATE) when working with continuous or high-dimensional instruments. By mapping complex instruments into a learned discrete representation, the authors apply Manski-style bounds while mitigating the instability that arises in adversarial or iterative training. Their two-stage procedure first estimates nuisance functions, then partitions the instrument space in a way that reduces variance in finite-sample settings. The key theoretical result is that these learned partitions yield valid, reasonably tight bounds for the CATE under the usual IV assumptions, but with fewer structural constraints than point-identification approaches. The proposed method is empirically evaluated on synthetic datasets.

## update after rebuttal

The authors' rebuttal addressed most of my concerns. However, I still view the work as somewhat incremental and, as such, I will maintain my borderline positive score (**3: Weak accept**).

**Claims And Evidence:**

The paper makes three primary contributions:

(1) a discrete representation approach that yields valid bounds on the CATE for complex Ivs

(2) a two-step estimation procedure claimed to be more stable than adversarial approaches

(3) theoretical guarantees that their learned bounds are valid under standard IV assumptions.

Contributions (1) and (3) are well-supported by evidence: detailed proofs and empirical results confirm that the proposed method produces narrower bounds compared to naive discretizations, and simulations demonstrate good alignment with the true CATE or oracle bounds. However, contribution (2) is not substantiated, as the authors neither provide direct comparative experiments nor theoretical justification for this claim. Thus, while the paper effectively motivates the stability advantage conceptually, it lacks sufficient evidence to show that this holds in theory/applications.

**Essential References Not Discussed:**

It could be beneficial to reference additional recent works on partial identification beyond Padh et al. (2023) and Levis et al. (2023) if they exist—for instance, studies that attempt to bound treatment effects with minimal assumptions in observational data, or new theoretical advances that might provide alternative bounding approaches. But otherwise, I think the related literature section is pretty comprehensive and includes the relevant references.

**Experimental Designs Or Analyses:**

The authors evaluate their method on simulated data, including scenarios with high-dimensional instruments (e.g., SNP-like variables) and known ground-truth oracles. I checked the design’s overall soundness—splits, metrics, and comparisons to baseline methods—and found nothing amiss. The coverage metrics and bound widths are computed in a consistent way for partial identification, and the analyses appear robust with no evident methodological flaws.

**Methods And Evaluation Criteria:**

The authors propose a two-stage approach: first, they train neural networks for nuisance functions (propensity and outcome regressions), then they learn discrete partitions of the instrument space to minimize bound width while balancing finite-sample variance. They evaluate performance primarily by coverage and average bound width, with additional metrics like MSD to gauge robustness under different partition sizes. These benchmarks are sensible for partial-identification methods in synthetic and semi-synthetic scenarios, though it would be valuable to see how the approach performs on a real-world dataset—even if we would only rely on qualitative assessments for validation.

**Other Comments Or Suggestions:**

*  I believe there is a typo in the definition of $\mathcal{L}_b$ in Eq (16) since he loss seems to decrease when the upper bound is lower than the lower bound which is not valid given that the objective is to minimize this loss.

* Additional intuition for the auxiliary loss $\mathcal{L}_{aux}$​ would be helpful, especially why cross-entropy is the best penalty to induce diverse partitioning.

**Other Strengths And Weaknesses:**

[Summary of the thoughts from the above sections]

Strengths

* The paper presents a straightforward method for applying Manski bounds to complex IV settings, supported by clear theoretical justifications.
* Empirical evaluations demonstrate effective coverage and tighter bounds in challenging, high-dimensional scenarios.
* The proposed method is modular, allowing the integration of different models for the first-stage nuisance estimation.

Weaknesses

* Although the authors claim improved stability over adversarial methods, no direct comparative experiments or formal proofs substantiate this claim.
* The benefits over naive clustering methods can be modest in simpler scenarios, with improvements becoming more pronounced in complex cases.
* While benchmarks presented are appropriate for partial-identification methods in synthetic and semi-synthetic contexts, it would be valuable to evaluate the method's performance on real-world datasets, even if such assessments were qualitative.

**Questions For Authors:**

[Summary of the thoughts from the above sections]

* Loss Mechanics: How does your loss function handle scenarios where the upper bound might dip below the lower bound, given that $\mathcal{L}_b$ is minimized directly?

* Stability Claim: You mention more stability than adversarial methods—could you expand on whether you attempted a direct comparison or if you can point to any theoretical rationale beyond avoiding min-max loops?

* Interpretation of Partitions: In a domain like genetics, do you see real-world interpretability in these learned clusters (e.g., discrete genotypes), or is it purely an internal mechanism?

* Auto-Tuning: Choosing the number of partitions $k$ seems crucial for balancing bound tightness and sample size per partition. Have you considered automatic selection methods (e.g., a validation-based approach)?
Overall, I consider this a strong partial-identification paper, and I support acceptance—pending clarifications on stability and the auxiliary loss.

Overall, I consider this a strong partial-identification paper, and I support acceptance—pending clarifications on stability and the auxiliary loss.

**Relation To Broader Scientific Literature:**

The approach addresses an existing gap in the literature where prior methods either required strong structural assumptions for point identification with IVs or relied on adversarial techniques to build valid bounds. The core idea—partitioning the instrument space and applying Manski-type bounds to the CATE function—is straightforward and heavily builds upon existing literature. Although not highly novel, the theoretical results and empirical validations are convincing, making this work a strong candidate for inclusion in the conference.

**Theoretical Claims:**

The theoretical claims are supported by detailed proofs, for which I commend the authors. I carefully reviewed Theorem 1 and Lemma 1 and they appeared correct to me. I skimmed the proof of Theorem 2, and it appeared correct as well, although I did not verify every detail thoroughly.

---

> ### Author Rebuttal · Authors · 2025-04-01
>
> Thank you for your positive and very actionable review! We took your comments at heart and improved our paper as follows.
>
> # Response to Claims and Evidence
>
> - **Stability of our method compared to adversarial approaches**: This is a very interesting point! As _widely investigated_ in the literature, adversarial (typically min/max) optimization suffers from _instability due to joint optimization of competing objectives_, which often leads to **slow convergence, gradient issues, and sensitivity to hyperparameters**. In contrast, our two-step approach decouples nuisance function estimation and partitioning to provide a more controlled, stable learning process. Even in simple **empirical** cases, when we estimated the bounds in an adversarial manner by retraining $\mu_\phi$ and $\pi_\phi$ after updating $\phi$ (instead of using our Eq. 3 and 4), we could _not even achieve convergence_. This highlights the superior stability of our method.
>
> **Action**: For the final version, we will report results with more exhaustive tuning of the adversarial baseline to provide a fair comparison.
>
> # Response to Methods and Evaluation Criteria
> - **Real-world data**: For benchmarking, we use synthetic DGPs that are **closely tailored to the real-world settings in Mendelian randomization (MR)**, such as polygenic risk scores and SNPs, and provide results for different levels of complexities. **We added new experiments with real-world data from a chemotherapy study containing genetic variants**. This allows us to the effect of exposure (smoking) on cancer progression (outcome).  **We provide the data description, results, and short interpretation here:** https://anonymous.4open.science/r/IVRep4PartId-714C/rebuttal/rebuttal_experiments.pdf. In sum, we observe that our method provides expected results while showing similar Width and **clearly more robust estimation** compared to the naive baseline, which confirms the effectiveness of our method.
>
> **Action:** We will include the real-world experiments in our paper.
>
>
> # Response to Weaknesses
> - Regarding the points “improved stability against adversarial methods” and “real-world experiments”, we kindly refer to our response from above.
> - **Limited benefits in simpler scenarios**: We agree that the benefits in terms of improved tightness are limited in the simpler setting. However, even though our method is designed for more complex settings, we still achieve similar performance in tightness (Width), while also providing more robust estimation (MSD) here. This shows that our method is suitable in both simpler and more complex settings.
>
> # Comments suggestions
> - **Loss formulation for $L_b$ in Eq. (16)**: While it might seem counterintuitive, our formulation in Eq. (16) is correct.  By our **Theorem 1**, we ensure that in population, the upper bound is always _greater than the lower bound_, and thus, the loss **cannot become negative**. In finite samples, by having large estimation errors in our bound estimates, this could change in theory. However, by using our **regularization loss** in Eq. (17), we directly avoid such unreliable estimations. Also, when running our experiments, we _never_ observed a negative loss, **undermining its applicability**.
> - **Intuition for $L_{aux}$**: Here, as a motivation for cross-entropy, we followed the heuristic that if the activations before the discretizations can be separated more easily, they should be more diverse. We also did some runs with the empirical Wasserstein distance loss to directly enforce more distributional distance in Z given the partitions. However, the cross-entropy loss performed stable. Further, the auxiliary loss only reduces the _estimation variance_, but does not affect the average _width or coverage_ (**see also our experiments in Appendix K**), thus turning out to be **useful but not crucial** for our method.
>
> # Response to questions
> - For loss mechanics and stability, we kindly refer to our answers above
> - **Interpretation of Partitions**: In principle, the clustering is only an internal mechanism to optimize our learning objective. However, under certain assumptions (see also **Appendix F.1**), maximizing for similar propensity scores within clusters and high heterogeneity between the clusters will lead to the closest bounds. Thus, one could implicitly identify genetic variations with similar influence on the exposure within one cluster.
> - **Auto-Tuning**:  Indeed a major benefit of our method compared to the baseline is the _robustness regarding $k$, since hyperparameter selection is hard in applied causal inference due to a _lack of access to the ground truth CATE_. However, we nevertheless provide practical guidelines in **Appendix F.2** for the optimal selection of $k$. Therein, we propose two approaches: (1) an expert-informed approach and (2) a data-driven approach, which can be used seamlessly in practice.

---

> > ### Comment · Reviewer_m2Ds · 2025-04-04
> >
> > Dear authors,
> >
> > Your rebuttal has addressed most of my concerns. However, I still view the work as somewhat incremental and, as such, I will maintain my score.

---

> > > ### Author Response · Authors · 2025-04-08
> > >
> > > Dear Reviewer m2Ds,
> > >
> > >
> > > Thank you for your response and your positive evaluation, we are happy that we could address most of your concerns! For a more detailed comparison of our contribution to prior work, and especially, for a detailed explanation why our novelty is orthogonal to the work of Schweisthal et al. (2024),  we kindly refer to our **rebuttal to reviewer xPaP**.
> > >
> > >
> > > Best regards,
> > >
> > > The Authors
> > >
> > >
> > > Schweisthal J, Frauen D, Van Der Schaar M, Feuerriegel S. “Meta-learners for partially-identified treatment effects across multiple environments.” ICML 2024.

---

### Official Review · Reviewer_XTJ8 · 2025-03-17

**Overall Recommendation:** 4

**Summary:**

The authors tackle the problem of partial identification of treatment effect with high dimensional instrument variables that have a complex relationship with the treatment. They do this by learning a discrete representation of the instrument variable and deriving a learning objective that does not require retraining the nuisance functions at each iteration step. A loss then introduced that includes extra regulariser terms that ensure good performance in the finite sample setting. They compare their method against a "naive" baseline that discretises the instrument (via kmeans) and applys existing bounds for discrete instruments. The performance of their estimator improves over this baseline.

## update after rebuttal

I will keep my positive score.

**Claims And Evidence:**

The claims of the paper are:
- Allow for partial identification with continuous, high dimensional instrument variables that may have complex relationship with the treatment.
- Introduce an algorithm that avoids adversarial learning.
- Demonstrate effectiveness both theoretically and empirically.

The claims are validated through the design of their algorithm as well as the good performance on simple and more complex tasks. It's a bit unclear how the theory shows that the algorithm is "effective", this contributions could be worded better.

**Essential References Not Discussed:**

Not to my knowledge.

**Experimental Designs Or Analyses:**

The experimental design makes sense for the problem at hand, increasing the complexity to show that the proposed changes are necessary.

**Methods And Evaluation Criteria:**

The paper compares on benchmarks with simple propensity and a more complex propensity to show that their method can gain performance in the first setting without losing performance in the first setting.

They introduce multiple metrics that tackle different properties of the bounds. However the MSE* and width* metric are only shown for Dataset 3.

**Other Comments Or Suggestions:**

- Figure 4: The writing in this figure is too small. The figure needs to be a lot larger. I would suggest even moving it to the appendix.
- Figure 3: This is not clear, is the training of the representation and updating of nuisance parameters not repeated multiple times? As is shown in Appendix H. The figure makes it seem like the representation is trained, then the nuisance updated which then results in the bounds.
- Motivation behind the metrics could be made clearer. What is a higher or lower value of a metric saying about the algorithm/performance under a decision?

**Other Strengths And Weaknesses:**

Strengths
- The paper tackles a well formed problem that is realistic and show that their solutions works.

Weaknesses:
- Some of the presentation could be improved.

**Questions For Authors:**

There does not seem to be much of an improvement on MSE* in dataset 3, which is the purported use case. Although you claim the the MSE* performance is better over the naive baseline, it seems to me that they are the same (within error). Am I missing something here? Why would the performance be so similar? Agains, the width is the same as well within error, and the coverage of the naive baseline is not much lower. How should I interpret these results?

Why do we care about the MSD score? In practice I will chose a single k and I want the performance for that k. If there is a procedure for choosing the best k, I would want to see the performance for that best k. For example, one method could be ok for all k, whereas another could be great, but only for a single k.

A clearer discussion of the metrics and the results would really help the narrative.

**Relation To Broader Scientific Literature:**

There is ample discussion of the difference in the settings compared to previous work. The fact that adversarial learning is not required is also a contibution.

**Theoretical Claims:**

The theoretical claims relate to the closed form computation of the nuisance functions, and the finite sample properties of the proposed algorithm. They seem correct.

---

> ### Author Rebuttal · Authors · 2025-04-01
>
> Thank you a lot for your positive review and your helpful feedback! We took your comments at heart and improved our paper as follows.
>
> # Response to Claims and Evidence
>
> **Wording of contribution**:  Thank you for this remark! We use the term “effective” to reference the good performance of our method when optimizing for our different objectives: (1) learning tight bounds with low average width, (2) providing valid bounds with full coverage, and (3) demonstrating robust estimation without requiring alternating learning or retraining of nuisance functions while providing stable performance with low MSD between different values of $k$. However, we agree that this terminology is a bit loose.
>
> **Action:** We will spell out more clearly which property of our method we refer to, and we will thereby spell out our contributions more clearly to avoid confusion.
>
>
>
> # Response to Methods and Evaluation Criteria
>
>
> **MSE*** **and Width*** : We agree that the MSE* and Width* would be also interesting metrics for Datasets 1 and 2. However, to calculate these metrics, we need to approximate the oracle bounds. Since we use continuous IVs for datasets 1 and 2 to show diverse settings, ground truth bounds cannot be estimated by closed-form solutions. For the high-dimensional Dataset 3 (and Dataset 4 in the additional experiments in **Appendix E**), we model the IVs to be dependent on a lower-dimensional discrete space such that we can approximate the oracle bounds. However, we agree that our paper would benefit from explaining this more thoroughly.
>
> **Action:** We will include an additional paragraph where we discuss our considered metrics in more detail.
>
> # Response to Weaknesses and Other Comments Or Suggestions
> - **Improved Presentation:** Thank you for your careful checks! **Action:** We will use the additional page of the camera-ready version to improve our presentation significantly including FIgures 3 and 4.
> - **Figure 3**: As you correctly pointed out and as presented in Appendix H, our current Figure 3 represents one training step. **Action:** To improve clarity, we will add our loss function to the final Figure and distinguish between (i) the training loop to update the parameters, and (ii) the final forward pass to estimate the final bounds.
> - **Motivation behind metrics**: As mentioned above, we will provide an additional discussion behind our choice of metrics. Here, we shortly want to summarize the most important aspects: The **Coverage**(# CATE within estimated bounds / n), and **Coverage*** (# ground truth bounds within estimated bounds / n)  should always be 1 and as high as possible to allow for reliable decision-making. The **Width** (sum of upper minus lower bound / n) and **Width*** (“Width for runs with Coverage* of at least 95%) express certainty about the CATE, and should be, in principle low. However, note that through estimation variance, low values of Width can become _falsely overconfident_ (tighter than the oracle bounds), which is why we introduce the Width* for Datasets 3 and 4. The **MSD** shows the variation for different values of $k$. Low values indicate robust behavior. This metric is especially important in real-world applications where we do not have access to the oracle CATE or oracle bounds, and thus cannot calculate the Coverage, Coverage*, or Width*. This makes it hard to select a $k$ which guarantees validity but also a certain tightness of the bounds. In **Appendix F**, we provide an extended discussion about the role of $k$ and practical guidelines for selection.
>
> # Response to Questions
>
> - **Results Dataset 3**: On average, **we show improvements of about 9 % for** **MSE*** **and 2% for Width***. For the latter, even though we filter for runs with coverage above 95%, there still can be up to 5% observations with overconfident bound estimations with low width for the naive baseline, biasing the gap to appear smaller. Further, the standard deviations are inflated by summarizing the runs over different $k$. Within $k$, these effects appear stronger, as shown in Figure 5. Here, we can also see that for low values of $k$, coverage is optimal for both methods while only for the rarer evaluated high values of $k$ does the coverage start to decline rapidly for the naive baseline.
> - **MSD score and role of $k$**: For the MSD score and the reporting of different $k$, we kindly refer to our answer from above. In **Appendix F**, we provide an extended discussion about $k$ and provide practical guidelines for selection that are suitable depending on the specific problem setting: (1) an expert-informed approach and (2) a data-driven approach. In our experiments, we provide the results for the respective values of $k$ in Table 3 and Figure 5, such that we can check the performance under different selection strategies.
>
> **Action: To address all points, we will add an extended discussion of the metrics and results to our paper.**

---

### Official Review · Reviewer_QCKo · 2025-03-21

**Overall Recommendation:** 2

**Summary:**

The paper proposes a method for partial identification, that is, bounding, of treatment effects in the instrumental variable (IV) setting. Specifically, the paper studies the scenario where the instruments are continuous and potentially high-dimensional. It proposes an approach for partial identification through a mapping of instruments to a discrete representation space, and learning the discrete representation by minimizing the width of the bounds. Theoretical results are provided claiming the validity of the bounds. Experiments are performed to show the effectiveness of the proposed method against a naive baseline.

## update after rebuttal
Thanks for the rebuttal to clarify a few things. Overall, the novelty of the paper is acceptable although somewhat incremental. The experimental results show that the proposed method is less sensitive to $k$ but are not convincing in showing that the proposed method is actually better than the naive baseline when a best $k$ is selected. I've updated my score to Weak Reject.

**Claims And Evidence:**

Overall, it feels that the paper exaggerates the novelty of the approach. It looks to me the proposed method is a somewhat direct extension of the existing closed-form bounds with discrete instruments. Rather than simply discretizing continuous instruments and using the existing bounds (the naive baseline), the paper proposes to learn a discrete representation of the continuous instruments that optimize bound width. The existing bounding results (Lemma 2) should be presented in the main paper.

In addition, I'm not sure about the claims regarding the "tightness" and "validity" of the bounds obtained by the proposed method throughout the paper. I think these claims are rather misleading.

-I don't understand the objective given in (1). To my understanding, we want valid and "tight" bound $b^-(x)$, $b^+(x)$ for each given $x$. Why would one want to minimize the "expected" bound width? In what sense are you claiming the "tightness" of the bounds given by (1)? Why do you call this "informative"? To my understanding, "informative" has a different meaning.

-Overall, I'm not sure about claiming tightness and validity under discretization and finite samples.

**Essential References Not Discussed:**

None.

**Experimental Designs Or Analyses:**

The overall experimental designs and analyses are reasonable. However, I have the following concerns:

-Can existing methods in IV settings with continuous treatments be tailored for binary treatment and therefore used as baselines (even if they are not directly tailored for binary treatment)?

-In Tables 1 and 2, some of the improvements in Width and MSE are small. Are the improvements actually statistically significant?

-In Table 3, it looks like the Naive baseline improves significantly with increasing $k$ values. What are the results for $k\geq 3$?

-I'm not sure it makes sense to show the results that are averaged over different $k$ values (what $k$ values are the results averaged over?). It makes more sense to me to select the best $k$ value. Have you tried choosing the best $k$ instead?

**Methods And Evaluation Criteria:**

Overall, the proposed method and evaluation criteria, including test datasets, are reasonable and make sense for the problem at hand.

**Other Comments Or Suggestions:**

-I believe assuming the causal structure in Fig. 1 implies the identifiability Assumptions 2 and 3.

-$s_1$, $s_2$ in (7) and (8) are not defined.

-I couldn't follow what the paragraph "Implications of Theorem 1" on page 4 is talking about.

-Eq. (24), (25) in Line 237 should be (3) and (4).

**Other Strengths And Weaknesses:**

none

**Questions For Authors:**

1. I don't understand the objective given in (1). To my understanding, we want valid and "tight" bound $b^-(x)$, $b^+(x)$ for each given $x$. Why would one want to minimize the "expected" bound width? In what sense are you claiming the "tightness" of the bounds given by (1)? Why do you call this "informative"? To my understanding, "informative" has a different meaning.

2. Can existing methods in IV settings with continuous treatments be tailored for binary treatment and therefore used as baselines (even if they are not directly tailored for binary treatment)?

3. I'm not sure it makes sense to show the results that are averaged over different $k$ values. It makes more sense to me to select the best $k$ value. Have you tried choosing the best $k$ instead?

**Relation To Broader Scientific Literature:**

To my understanding, the work directly extends the existing closed-form bounds with discrete instruments given in Lemma 2.

**Theoretical Claims:**

I did not check the details of the proofs for theoretical claims. They look reasonable. Theorem 1 is a somewhat direct extension of existing results. Lemma 1 is standard.

---

> ### Author Rebuttal · Authors · 2025-04-01
>
> Thank you for your review and the opportunity to clarify multiple points of our paper! We took your comments at heart and improved our paper as follows.
>
> # Response to concerns around claims and evidence, theoretical claims, and relation to broader scientific literature
>
> - **Novelty of our paper and implications of Theorem 1**:  We would like to emphasize that, while the extension of existing closed-form solutions to complex instruments is novel (i.e., even the naive baseline has not been considered yet), this is only _a small part_ of our contribution. Importantly, instead of **only deriving** valid bounds, we focus on a method for **improved estimation**. Based on this, we introduce the new objective of directly minimizing bound width during representation learning (Eq. 8).
>
> Importantly, in Theorem 1, we show that this objective can be optimized _without alternating learning_: we can directly evaluate the necessary quantities from pre-trained nuisance functions by leveraging Eq. 3 and Eq. 4. (otherwise, we would need to retrain $\mu_\phi$ and $\pi_\phi$ after every update step of $\phi$ in an alternating way). Thus, with our formulation, **we avoid instability and high computational cost**. (see also “Implications of Theorem 1” in our paper). Lastly, to ensure robustness in finite samples, we regularize for balanced partition probabilities (Eq. 14), _reducing estimation variance as shown in our Theorem 2._
>
> - **Clarifications on our objective in Eq. (1), and claims about “tightness” and “validity”**: The validity of our bounds in population follows straightforwardly from Theorem 1. In finite samples, as for every method, full validity (i.e., guaranteed 100% coverage) can never be proven. However, by using our regularization loss, we mitigate the risk of overconfident estimation, which is unlike existing methods that do not account for estimation uncertainty.
>
> In addition, we refer to tightness by minimizing the expected bound width as much as possible given some representation $\phi$. Importantly, this usually does not lead to **sharp bounds** (i.e., the “tightest possible ones” for all $x$ which would only minimize the population tightness in Eq. 12), since this would not allow us to reduce the estimation variance. This is also our motivation for our objective in Eq. (1): _We aim to reduce the average bound width to get robust estimates of “informative” bounds for the largest part of the population_. Here, “informative” refers to “being useful for the underlying decision-making problem”. E.g., for classical CATE estimation, informative means that the lower bound is greater or the upper bound is below some decision threshold (often 0).
>
> **Action:** While we are consistent in our terminology with previous work on partial identification, we will add a discussion about the different terms to improve the clarity of our work.
>
> # Response to experimental designs and analyses
>
> - **Adaptation of existing methods**: As these methods often include additional assumptions over the continuous interventional distribution in the form of additional constraints, the adaptation is not straightforward. However, in **Appendix E**, we provide results for additional adapted baselines, such as extensions of methods that were designed for point identification.
> - **Significance of results**: As standard in machine learning, we avoid making inferences about statistical significance since this would require careful adjustment for dependencies and multiple testing, but instead report mean and standard deviation over multiple seeds.
> - **Role of $k$ in Table 3**: Dataset 1 is defined in a simple way such that a naive partitioning with $k=3$ can already closely approximate optimal bounds. Interestingly, with k=4 the partitioning gets worse, yielding only a coverage of 0.49 for the naive baseline, while still yielding 1.00 for our method. _This shows the robustness of our method_, and even may indicate that the narrow bound width of 0.83 for the baseline for k=3 may be due to _falsely overconfident_ bound estimates.
> - **Evaluation and selection of $k$**: We average over the k-values considered in our sensitivity analysis as reported in Table 3 and Figure 5. The intuition here is that, in real-world applications, one does not have access to the ground-truth CATE or oracle bounds, and thus coverage cannot be checked to select the best k. When only considering width as the criterion, this would lead to always choosing the tightest bounds, which are likely to not yield full coverage, especially for the baseline. In **Appendix F**, we provide an extended discussion over the role of $k$ and give guidelines on how to select k in practice.
>
> # Response to other comments or suggestions
> - Thanks for your careful remarks! We will adjust our paper accordingly. Regarding the “Implications of Theorem 1”, we kindly refer to our answers above.
>
> # Response to Questions
> - Thank you for your questions! We addressed all of them in our answers above.

---

> > ### Comment · Reviewer_QCKo · 2025-04-04
> >
> > Thanks for your rebuttal.
> >
> > -The novelty of the paper is incremental but acceptable.
> >
> > -To my understanding, the goal of partial identification is to find valid and sharp/tight bound $b^-(x)$, $b^+(x)$ for each given $x$. So, the ideal objective should not be Eq. (1), which is already an approximation or proxy of the partial identification problem. What space are you optimizing in (1)? On the other hand, it makes sense to learn a discrete representation that minimizes the expected bound width. However, claiming learning "tight" bound as a contribution throughout the paper is misleading when "tightness" here actually means ``minimizing the expected bound width as much as possible given some representation $\phi$'', a measure specific to the problem the paper defined.
> >
> > -Comparison with the naive baseline and the role of $k$: The experimental results indeed show that the proposed method is less sensitive to $k$ than the naive baseline, that is, more robust. But, at the end of the day, one uses the algorithm to output a bound based on some $k$. It doesn't make sense to compare the performances of the two algorithms based on the average over different $k$ values. Other than being less sensitive to $k$, it's hard to draw the conclusion that the proposed method is actually better than the naive baseline based on the experimental results shown. In the complex Dataset 3, width* and MSE* are actually comparable.

---

> > > ### Author Response · Authors · 2025-04-08
> > >
> > > Thank you very much for your response and for giving us the chance to elaborate on the remaining points of concern! We are confident we can address all of them with minor changes in our paper.
> > >
> > > - We are happy that we were able to address some of your concerns regarding our novelty. For a more detailed comparison of our contribution to prior work, we kindly refer to our **rebuttal to reviewer xPaP**.
> > >
> > > - We understand that our objective in Eq. (1) differs from the typical formulation in some other partial identification papers, but it targets the same objective. Thus, we would like to show the connection to the more traditional formulation. First, we can formulate the space we are optimizing over as the
> > > compatible distributions (distributions over joint data including $U$, unobserved) that are compatible with the observed data distribution
> > >
> > > $$
> > > \mathcal{M} = \{ \mathbb{P}^*(z, a, x, u, y) \mid \mathbb{P}(z, a, x, y) = \int \mathbb{P}^*(z, a, x, u, y)  du \}
> > > $$
> > >
> > >
> > >
> > > **Option A: classical formulation**:
> > >
> > > Here, recent literature often formulates the goal of partial identification such that
> > > $$
> > >      b^+(x) =  \sup_{\mathbb{P}^\ast \in \mathcal{M}}  \tau_{\mathbb{P}^\ast}(x)
> > > $$
> > > (and equivalently for $b^-(x)$)
> > > gives optimal _sharp_ bounds.
> > >
> > > In our paper, we make use of another formulation:
> > >
> > > **Option B (ours)**
> > >
> > > To get _valid bounds_, we define the set
> > >
> > > $$
> > > \mathcal{V}_{+} = \{ b : \mathcal{X} \to \mathbb{R} \mid \tau_P\ast(x)  \leq b(x) \text{ for all } P^* \in \mathcal{M},\ x \in \mathcal{X} \}
> > > $$
> > >
> > > (and equivalently for $\mathcal{V}_{-} $).
> > >
> > >
> > > Then, we can minimize the bound width via
> > >
> > > $$
> > > b_*^-,\ b_*^+ \in \arg\min_{b^- \in \mathcal{V}-,b^+ \in \mathcal{V}+} \mathbb{E}_X [b^+(X) - b^-(X)]
> > > $$
> > >
> > >
> > > This is **equivalent** to option A and also gives _sharp_ bounds (i.e., if the width for every $x$  is minimal, then the expected width will be minimal, too). Only later, when we restrict the bounds to functions that can be expressed dependent on the discretization $\phi$, we don’t necessarily yield _sharp_ bounds, but instead, focus on robust estimation.
> > >
> > >
> > > We use Option B because we have a good notion of what valid bounds are from Theorem 1. Here, we can directly incorporate the objective above into our loss function as the _bound width minimization_ term. Further, while our usage of the term “tight” is motivated by the formulation above (minimizing the expected width), we fully agree that this contains ambiguity and can lead to confusion about the distinction to the term “sharp”.
> > >
> > >
> > > **Action:** We will add both formulations and the motivation for using Option B from above to Sec. 3  to improve the clarity of our paper. Further, we will update our usage of the term “tight” and instead directly refer to “reducing the expected bound width” to dissolve ambiguities.
> > >
> > >
> > > - In our experiments, we report the results averaged over multiple $k$ because, as usual in causal inference, hyperparameter tuning is more challenging without access to the ground truth CATE, and thus there are different strategies to select $k$ (see also **Appendix F.2**). Thus, taking the average over the reasonably selected $k$ can be seen as reporting the summarized performance over different strategies that would have resulted in selecting the different values of $k$ (e.g., by the expert-informed approach). However, we agree that this presentation is not optimal, and even leads to _less clear_ performance gains, as when considering $k$ separately (see Figure 5, for more details we kindly refer to our **rebuttal to reviewer XTJ8, “Response to Questions”**. Further, in our updated Figure 5 (https://anonymous.4open.science/r/IVRep4PartId-714C/rebuttal/sensitivity_k_combined.pdf), we now also report the MSE* over the different $k$ runs, showing clearer improvements than the averaged results).
> > >
> > >
> > > Thus, **we now report the results for the dataset using our data-driven approach** for selecting $k$, which (without access to ground truth CATE) results in a selection of $k$ = 15.
> > >
> > > | **Metric**                  | Naïve              | **Ours**            | **Rel. Improve** |
> > > |----------------------------|--------------------|---------------------|------------------|
> > > | Coverage* [↑]          | 0.600 ± 0.547        | **0.937 ± 0.057**     | **56.17%**         |
> > > | Width* [↓]            | 1.818 ± 0.076        | **1.788 ± 0.012**     | **1.65%**         |
> > > | MSE* [↓]               | 0.094 ± 0.030      | **0.085 ± 0.009**     | **9.57%**         |
> > >
> > > We observe that, while we have some smaller gains in Width* and MSE*, the Coverage* of the naive baseline is low while ours remains high, **showing the key benefit of our method for reliable decision-making**.
> > >
> > > **Action:** We will update our results as shown above and move current tables averaging over $k$ to the Appendix to improve the presentation of our paper and better highlight the strength of our method.

---

### Official Review · Reviewer_d3FE · 2025-03-24

**Overall Recommendation:** 3

**Summary:**

To partially identify and estimate the bounds of the conditional average treatment effect (CATE) with potentially high-dimensional instruments, the authors propose a two-step approach. This method first learns discrete representations of the complex instrumental variables $Z$, then derives closed-form bounds based on these representations, and finally maps the bounds back to the original problem. The authors present two theorems for computing these bounds: one from a population perspective and the other from a finite-sample perspective. Additionally, they introduce a neural approach for learning CATE bounds when dealing with complex instruments. Experimental results demonstrate that the proposed method achieves superior validity and robustness.

**Claims And Evidence:**

Yes. I did not find any problematic claims.

**Essential References Not Discussed:**

None.

**Experimental Designs Or Analyses:**

The authors simulated three datasets to conduct MR simulations. The evaluation metrics include coverage, width, and MSD for Datasets 1 and 2, while for Dataset 3, they introduced three analogous metrics. Additionally, the authors provided implementation details and conducted a sensitivity analysis on different $k$. However, I believe the comparison is limited to only a naive baseline, lacking comparisons with more alternative methods. Furthermore, the experimental settings could be adjusted to settings where existing methods are applicable, facilitating a fairer comparison.

**Methods And Evaluation Criteria:**

Yes, the proposed methods and evaluation criteria are well-aligned with the problem at hand. The method effectively captures the key characteristics of CATE by complex $Z$. The evaluation metrics such as convergence, width, and MSD effectively illustrate the performance of the proposed methods. However, the benchmark dataset used in the study is a simulated dataset, while it is better to perform experiments on a real-world dataset.

**Other Comments Or Suggestions:**

1.	The sentence 'There are many reasons, including costs …' reads awkwardly and contains a grammatical issue.

2.	Some notations in Theorem 2, such as $q$ and $\theta$, only appear in the proof and are not found in the main text, which may
confuse readers.

3.	The abbreviation "wrt." should be explained the first time it appears.

4.	If possible, it would be helpful to place Figure 4 near Figure 5.

**Other Strengths And Weaknesses:**

- The paper is well-written and clearly structured, making it easy to follow the key arguments and contributions. The figures and illustrations are well-designed and effectively convey the key insights of the paper.
- The contribution in this paper appears to be somewhat limited. The core contribution primarily involves redefining the forms of $\mu$ and $\pi$ from Lemma 2 under a discrete representation $\phi$, directly leading to Theorem 1.
- In Table 3, the naïve method for dataset 1 performs significantly better in terms of width. Can the authors provide an explanation for this observation?
- It seems that $k$ is important for the performances, from the sensitivity analysis. Performances are different with different $k$. How to set $k$ in practice?

**Questions For Authors:**

- The experimental section lacks real-world datasets and relies solely on simulated experiments, which do not demonstrate the performance of the method in real-world settings.
- Additionally, the comparison with other methods is insufficient. Since the authors claim that other methods are not suitable for their setting, and their method can handle more complex instruments $Z$, why not apply their method in a simpler setting and compare it with other methods? This would make the argument more convincing.
- There are many invalid IVs in MR. In this case, does the proposed method still work well? Or how to deal with these invalid IVs?

**Relation To Broader Scientific Literature:**

This paper introduces a two-stage approach for partial identification of treatment effects using high-dimensional instruments, leveraging discrete representations to obtain closed-form bounds.
Previous works on Machine Learning for CATE estimation (e.g., Singh et al., 2019) focused on settings where the treatment effect can be identified and used ML methods with favorable properties (e.g., Kennedy et al., 2019).
In partial identification with binary treatment, Robin and Manski proposed closed-form ATE bounds for bounded $Y$, and in the binary case, Balke and Pearl derived tighter bounds.
For discrete variables, some researchers have provided broad overviews, but the issue is that these methods fail to effectively leverage continuous and high-dimensional instrumental variables to learn tighter bounds. In contrast, the method proposed by the authors expands the applicable scope and achieves tighter bounds.

**Theoretical Claims:**

I checked Theorem 2 and found no discussed issues.

---

> ### Author Rebuttal · Authors · 2025-04-01
>
> Thank you for your detailed feedback and the overall positive evaluation of our manuscript! We will take all your comments to heart and improve our manuscript accordingly. Below, we provide answers to all your questions
>
> # Response to Methods and Evaluation Criteria
>
> - **Benchmark datasets**: We agree that providing insights on real-world data is an interesting extension for our paper. However, it is standard in causal inference literature to rely on _synthetic data-generating_ processes for benchmarking, since the CATE (and for our setup, also optimal bounds on the CATE) can **never be observed in real-world data**. Thus, we provide benchmarking using synthetic DGPs that are **closely tailored to the real-world settings in Mendelian randomization (MR)**, such as polygenic risk scores and SNPs, and provide results for different levels of complexities. Further, by using our DGP for datasets 3 (and 4 in the Appendix), we can approximate _oracle bounds_, allowing us to check for validity also with respect to optimal bounds and not only compared to the CATE, **which is an extension over previous work** in similar settings.
>
> **Action**: To further strengthen our paper, we provide **additional insights on real world data** in MR (https://anonymous.4open.science/r/IVRep4PartId-714C/rebuttal/rebuttal_experiments.pdf). **Overall, our findings are consistent with our synthetic setup and indicate robust performance of our method**
>
> # Response to Experimental Designs or Analyses
>
> - **Additional baselines and experimental setups**: Thank you! Importantly, we focus on the highly relevant setting with _complex instruments_ and _binary treatments_ as it has been neglected in prior work. On the other hand, this implies, that our method is _not tailored or applicable_ to setups considered by existing papers (i.e., our method cannot be naively adopted to continuous treatments; and, for simple discrete IVs, no tailored representation learning is necessary since the closed form bounds can be estimated directly). Thus, we focus on our considered datasets. However, we agree that additional baselines besides the naive one are interesting. Thus, in **Appendix E**, we provide results for additional adapted baselines to show that our tailored method is clearly superior and necessary.
>
> # Response to Strengths and Weaknesses
>
> - **Contribution**: We agree that the reformulation of $\mu$ and $\pi$ is one of our main contributions as this allows us to avoid alternating learning. However, we would kindly highlight our major other contributions: (i) **Novel setting**: We are the **first** to directly focus on **partial identification with complex instruments such as in MR**, (ii) **Our learning algorithm**: We do _not_ only avoid alternating learning and leverage discrete representation learning but also encourage reduced estimation variance by our _tailored loss function_ as shown in Theorem 2.
>
> - **Dataset 1, Table 3 performance**: As the DGP for Dataset 1 is simple, a naive clustering with $k=3$ could already lead to close-to-optimal bounds. However, importantly, for Datasets 1 and 2, _lower width does not necessarily indicate better performance_. By using only finite data for estimation, the bounds could also be **falsely overconfident**, even if they lead to full coverage of the true CATE, as the naive baseline does not try to reduce estimation variance. Since, in these DGPs, we use _continuous IVs_, we cannot approximate the oracle bounds (as for Datasets 3 and 4). Instead, the main finding here is that **our method leads to robust estimates for different $k$, while the naive baseline highly depends on the selection of $k$**.
>
> - **How to set $k$ in practice**: As mentioned, a major benefit of our method compared to the baseline is the _robustness regarding $k$_. However, since hyperparameter selection is hard in applied causal inference due to a _lack of access to the ground truth CATE_, we provide practical guidelines in **Appendix F2**. Therein, we propose two approaches: (1) an expert-informed approach and (2) a data-driven approach, which can be used seamlessly in practice.
>
> # Response to other Comments
>
> Thank you for your careful reading! All of your points are very helpful and we will include them in the camera-ready version of our paper.
>
> # Response to Questions
> - Thank you for your questions! For the first two questions, we kindly refer to our responses above.
> - **Invalid IVs**:  As in typical MR and IV settings, we need to ensure that the exclusion and independence assumptions hold. However, unlike usual methods, we do **not** require the relevance assumption: we can – in principle – also use SNPs that are not associated with the exposure /treatment. Since these irrelevant IVs do not lead to closer bounds, our method will just not use their information. Our datasets 3 and 4 indeed contain 75% of irrelevant IVs, which demonstrates the **strong performance of our method in this relevant scenario**.

---

### Decision · Program_Chairs · 2025-05-01

**Decision:**

Accept (poster)

**Comment:**

This paper provides a methodology and analysis for estimating (potentially) high-dimensional Conditional Average Treatment Effects (CATE) under partial identification. As a means of identification, the authors derive bounds with the discretized instrumental variables and analyze the associated estimators and their asymptotic properties. A neural approach for estimating these bounds is also presented.

The paper is clearly written and accessible, and it has been positively received overall. The comprehensiveness of the experimental setup is another strength. However, as noted by several reviewers—including expert reviewers—the technical contributions of this paper are rather limited. As a result, the review scores are divided.

The area chair has studied this issue in depth and concurs that the main results are somewhat similar to existing work. Although the paper does offer several additional contributions, the limited novelty of the core results diminishes the overall impact of these additional elements. Nonetheless, the quality of the paper itself is recognized.